# Associative Memory augmented Asynchronous spatiotemporal Representation learning for Event-based perception

**Uday Kamal**[*], **Saurabh Dash**[*][†]**& Saibal Mukhopadhyay**
School of Electrical and Computer Engineering
Georgia Institute of Technology
{ukamal6,saurabhdash}@gatech.edu, saibal@ece.gatech.edu

## Abstract

We propose *EventFormer*– a computationally efficient event-based representation learning framework for asynchronously processing event camera data. Event-Former treats sparse input events as a spatially unordered set and models their spatial interactions using self-attention mechanism. An associative memory-augmented recurrent module is used to correlate with the stored representation computed from past events. A memory addressing mechanism is proposed to store and retrieve the latent states only *where* these events occur and update them only *when* they occur. The representation learning shift from input space to the latent memory space resulting in reduced computation cost for processing each event. We show that EventFormer achieves $0.5\%$ and $9\%$ better accuracy with $30000\times$ and $200\times$ less computation compared to the state-of-the-art dense and event-based method, respectively, on event-based object recognition datasets.

## 1 Introduction

Ultra-low power, high dynamic range ($> 120dB$), high temporal resolution, and low latency makes event-based cameras (Brandli et al., 2014; Suh et al., 2020; Son et al., 2017; Finateu et al., 2020) attractive for real-time machine vision applications such as robotics and autonomous driving (Falanga et al., 2020; Hagenaars et al., 2020; Sun et al., 2021; Zhu et al., 2018; Gehrig et al., 2021). Convolutional and recurrent neural network-based methods, originally developed for frame-based cameras, have demonstrated good perception accuracy on event camera (Gehrig et al., 2019; Baldwin et al., 2022; Cannici et al., 2020b). But they rely on temporal aggregation of the events to create a frame-like dense representation as input thereby discarding the inherent sparsity of event data and resulting in high computational cost (Figure 2). Recent works have explored event-based processing methods for object recognition to exploit data sparsity. Examples of such methods include Time-surface based representation relying on hand-crafted features (Lagorce et al., 2016; Sironi et al., 2018; Ramesh et al., 2019), 3D space-time event-cloud (Wang et al., 2019), and Graph-based methods (Li et al., 2021c; Schaefer et al., 2022). These methods adopt event-based processing to achieve lower computational costs but do not achieve similar performance compared to the dense-representation based methods (Figure 2). This necessitates computationally efficient algorithms that exploit sparsity and achieve high accuracy.

We propose an associative memory-augmented asynchronous representation learning framework for event-based perception, hereafter referred to as *EventFormer*, that enables computationally efficient event-based processing with high performance (Figure 1). As events are triggered asynchronously, an event-based processing algorithm must generate and maintain a higher-order representation from the events, and efficiently update that representation to correlate a new event with the past events across space and time. One way to address this is to include a recurrent module at each pixel to track history of past events (Cannici et al., 2020a). However, the associated processing and memory requirement of such a method increases exponentially with the number of pixels. Motivated by recent works in memory augmented neural networks(Kumar et al., 2016; Ma et al.,

---

[*]equal contribution     [†]Author now at CohereAI

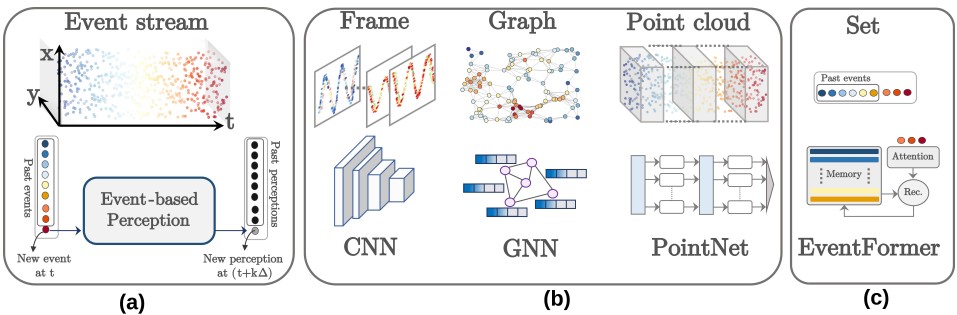

Figure 1: **Comparison with existing works. (a)** Event-based perception algorithms where perception latency $k\Delta$ is proportional to the event-generation rate($\Delta$). **(b)** Existing works include (left) *dense processing* that aggregates events into a frame and use CNN, *event-based processing* including (mid) GNN on spatiotemporal event-graph, or (right) PointNet like architectures treating events as point-cloud. These methods either re-process (frame and point cloud) or store the past events (graph) to spatiotemporally correlate with new events. **(c)** EventFormer encodes the spatiotemporal interaction of the past events into a compact latent memory space to efficiently retrieve and correlate with the new events without requiring to store or re-process the past events.

2018; Karunaratne et al., 2021), we address the preceding challenge by maintaining a spatiotemporal representation associated with past events, occurring at various pixels, as the hidden states of an Associative Memory. As learning spatiotemporal correlation is shifted from the high-dimension input (pixel) space to a compact latent memory space, EventFormer requires an order of magnitude lower floating point operations (FLOPs) per event to update the memory. To the best of our knowledge, EventFormer is the first associative memory-augmented spatiotemporal representation learning method for event-based perception. The key contributions of this paper are:

- EventFormer maps the spatiotemporal representation of the incoming event stream into the hidden states of an associative memory and uses a lightweight perception head directly operating on these states to generate perception decisions.

- The spatiotemporal representation update mechanism activates only 'when' and only 'where' there is a new event using unstructured set-based processing without requiring to store and re-process the past events.

- We propose a new query-key association-based memory access mechanism to enable spatial location-aware memory access to retrieve the past states of the current event locations.

Given a new event (or a set of events), our model generates a spatial representation by computing their spatial interaction through self-attention mechanism (Vaswani et al., 2017)and retrieves the past spatiotemporal states related to that pixel location(s) from the memory. The retrieval process exploits a novel location-based query and content-based key-value association mechanism to extract correlations among the past events at neighboring pixels. Once we get the past states, along with our present spatial representation, a recurrent module takes them as input and generates the *refined* state information. The associated hidden states of the memory are updated with the new information. We evaluate EventFormer on object recognition task from event-camera data. In our experiments on existing benchmark datasets, N-Caltech101 (Orchard et al., 2015) and N-Cars (Sironi et al., 2018), EventFormer shows an excellent computational advantage over the existing methods (both dense and event-based) while achieving performance comparable to the dense methods (Figure 2).

**Related Work:** *Dense methods* convert events to dense frame-like representation and process them with standard deep learning models such as CNNs (Maqueda et al., 2018; Gehrig et al., 2019; Cannici et al., 2020a). These methods are synchronous in nature as they generate dense inputs by binning and aggregating events in time and generate output only when the entire bin is processed. *Event-based methods* update their representations with a new event and generate new output. Methods such as (Lagorce et al., 2016; Sironi et al., 2018; Ramesh et al., 2019) compute time-ordered representation (also known as time-surface) in an event-based manner with fewer computations. However, their reliance on fixed, hand-tuned representation results in sub-optimal perfor-

mance compared to the data-driven methods (Gehrig et al., 2019). Recently, graph-based methods (Messikommer et al., 2020; Li et al., 2021b; Schaefer et al., 2022) have been considered that generate a spatiotemporal event graph and process them with GNN. For a new event, the graph is updated by removing nodes associated with past events and adding new ones. As these methods require subsampling the events to reduce the size of the graph (and hence, computation), performance is reduced as well. Also, they need to store the past events inside the graph to correlate with the new event causing additional computation and memory overhead. *Point cloud*-based methods can inherently process unstructured event camera data in a permutation invariant manner (Wang et al., 2019; Vemprala et al., 2021). However, they process over a sliding window of time $\tau$ and need to re-process the past events for every new event to establish correlations leading to redundant computations. *Memory Augmented Neural Networks* typically combine an external associative memory with a recurrent controller (Kumar et al., 2016; Ma et al., 2018; Karunaratne et al., 2021) resulting in enhanced memory capacity compared to the vanilla recurrent module (Cho et al., 2014). Cannici et al. (2020) propose a pixel-wise memory-based representation learning for event-camera data that stores states at each pixel location (Cannici et al., 2020a). However, it requires a computationally expensive feature extractor as spatiotemporal correlations among these states are not considered. Event camera has also been applied for more complex tasks including egomotion, motion segmentation(Parameshwara et al., 2021b), and depth prediction (Hidalgo-Carrió et al., 2020). To broadly categorize, these methods either fall into *dense frame-based* representation where classical image processing (Mitrokhin et al., 2018) and end-to-end learning-based methods (Zhu et al., 2019; Mitrokhin et al., 2019) have been applied or *3D-point cloud representation* where GNNs has been adopted(Mitrokhin et al., 2020) to learn to perform these dense prediction tasks. While most of the existing works in this direction adopt a dense processing-based encoder-decoder structure, a spike-based asynchronous formulation has also been explored in (Parameshwara et al., 2021a).

## 2 METHOD

**Preliminaries and Problem Formulation:** An event-based camera consists of sensors at every pixel location that can respond to the change of brightness asynchronously and independently (Gallego et al., 2020). Mathematically, a set of events produced at pixel locations $(x, y)_i$ ($0 \leq x \leq W, 0 \leq y \leq H$) during a time interval $\tau$ can be defined as a time-ordered sequence, $\mathcal{E}_\tau = \{(x_i, y_i, t_i, p_i) \mid t_i \in \tau, T = \sup_i t_i\}$ where $t_i$ denotes the event triggering timestamps, $p_i \in \{-1, 1\}$ as the polarity (relative change in brightness), and $T$ as the total time of observation. In this work, we consider the spatial locations of the events as the input, that is: $\mathcal{E}_\tau = \{(x_i, y_i)\}$. Our goal is to learn a compute efficient parametric mapping $\mathcal{F} : \mathcal{E}_\tau \rightarrow \mathcal{M}_{\mathcal{E}_\tau}$ that can convert the raw event sequence $\mathcal{E}_\tau$ into a suitable representation $\mathcal{M}_{\mathcal{E}_\tau} \in \mathbb{R}^{m \times d}$ (where $m \times d$ defines the dimension of the feature space) with event-based processing capability. This implies that $\mathcal{F}$ needs to have the capability to update its representation as soon as a new event(or a new event sequence) arrives without storing or recomputing the past events.

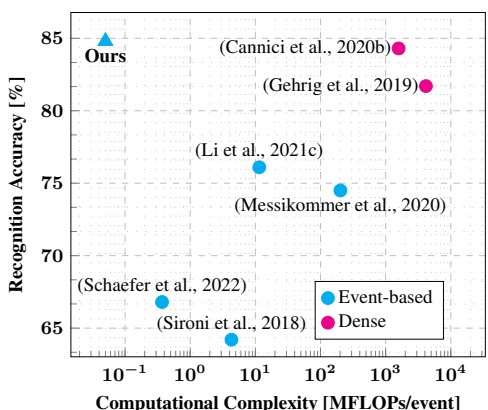

Figure 2: **Recognition performance vs computation.** On N-Caltech101 dataset, EventFormer has about 30000× lower compute cost than the SoTA dense method (Cannici et al., 2020b) with 9% improvement in accuracy compared to the best performing event-based method (Li et al., 2021c).

**Overview of the Framework:** Eventformer consists of a positional encoder (Li et al., 2021a) followed by a pairwise interaction module to compute the spatial interactions among the events (Figure 3). A recurrent module takes this output, $\mathcal{Z}_t$ as the current input state, and computes the spatiotemporal representation $\mathcal{X}_t$ by retrieving the past hidden states $\mathcal{H}_{t-1}$ stored in an associative memory, $\mathcal{M}$. The output of this recurrent module is used to update the memory representation $\mathcal{M}_t$, which

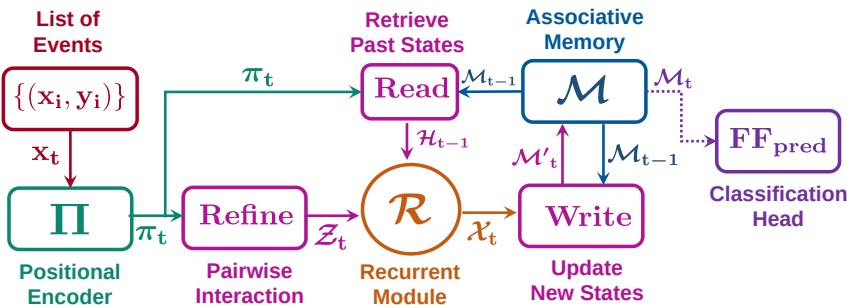

Figure 3: **EventFormer architecture.** EventFormer generates positional embedding $\pi_t$ from a list of new events $x_t$ at time $t$, and calculates their pairwise interaction, $\mathcal{Z}_t$ using self-attention. A recurrent unit extracts the past states $\mathcal{H}_{t-1}$ associated with $x_t$ from an associative memory and computes $\mathcal{X}_t$ as the current states. This is further encoded into the latent memory space, $\mathcal{M}_t$ to process future events. A linear layer uses $\mathcal{M}_t$ to predict the target class.

---

**Algorithm 1: Compute the event-based representation through EventFormer**

**Input:** A list of $n$ 2-dimensional pixel coordinates of the events at time $t$:
$$x_t = \{(x_i, y_i) \mid i \leq n\}$$
**Output:** Prediction vector $Y_t \in \mathbb{R}^c$ where $c$ denotes the number of classes.
**Hyperparameter**: The dimension of the representation vector $d$, row size of the memory $m$, number of stacks for the `Refine` operator $R$.
Initialize the memory representation with the learned initializer, $\mathcal{M}_0 \in \mathbb{R}^{m \times d}$

1   $\pi_t \leftarrow \Pi(x_t) := \frac{1}{\sqrt{d}} \left( \cos(x_t W_p^T) \,\|\, \sin(x_t W_p^T) \right)$;
2   $\mathcal{H}_{t-1} \leftarrow \texttt{Read}(\pi_t, \mathcal{M}_{t-1})$;          ▷ *Read the past memory-representation*
3   **foreach** $r \in R$ **do**
4      $\pi_t \leftarrow \texttt{Refine}_r(\pi_t, \pi_t)$;            ▷ *Compute event interactions*
5      $\mathcal{Z}_t \leftarrow \pi_t$;
6   $\mathcal{X}_t \leftarrow \mathcal{R}(\mathcal{Z}_t\,,\, \mathcal{H}_{t-1})$;             ▷ *Compute the current states*
7   $\mathcal{M}'_t \leftarrow \texttt{Write}(\mathcal{M}_{t-1}, \mathcal{X}_t)$;      ▷ *Compute new memory representation*
8   $\alpha_t \leftarrow \texttt{sigmoid}(\texttt{Erase}(\mathcal{M}_{t-1}, \mathcal{X}_t))$;      ▷ *Compute the update factor*
9   $\mathcal{M}_t = \alpha_t \mathcal{M}_{t-1} + (1 - \alpha_t)\mathcal{M}'_t$;         ▷ *Update the memory*
10   $\mathcal{M}_t^{flat} \leftarrow$ Reshape $\mathcal{M}_t$ into a 1-dimensional vector;
11   $Y_t \leftarrow \texttt{FF}_{\texttt{pred}}(\mathcal{M}_t^{flat})$;             ▷ *Prediction at time $t$*
12   **return** $Y_t$.

---

is used by a classification head for the recognition task. To facilitate these operations, EventFormer has three unique `Operators`: `Read` to retrieve the past representation, `Write/Erase` to update the memory with new representation, and, `Refine` to compute the spatial correlation among the events. Both `Read` and `Write/Erase` operators use multihead residual attention (Vaswani et al., 2017) as a building block. The `Refine` operator adopts an efficient version of this building block (Shen et al., 2021) to address the quadratic memory and compute requirement (that can become computationally prohibitive for a very fast-moving object generating a large number of events at a given time) of the traditional dot-product attention mechanism (Vaswani et al., 2017). Algorithm 1 shows an implementation of the overall Eventformer architecture.

***Operator details:*** An $\texttt{Operator}(\mathcal{A}, \mathcal{B})$ takes $\mathcal{A} \in \mathbb{R}^{n \times d}$ and $\mathcal{B} \in \mathbb{R}^{m \times d}$ as input and maps them into the $Q$ (query), $K$ (key), and $V$ (value) space using a linear transformation. The attention mechanism maps the $Q$ to outputs as follows:

$$\texttt{Attn}(Q, K, V; a) = a(QK^\top)V \tag{1}$$

$QK^\top \in \mathbb{R}^{n \times m}$ measures the pairwise similarity of the query and key vectors. $a(QK^\top)V \in \mathbb{R}^{n \times d}$ is a weighted sum of the value vectors where the weights are computed using scaled softmax activation $a(.) = \texttt{softmax}(./\sqrt{d})$. The multihead residual attention computes multiple attention by

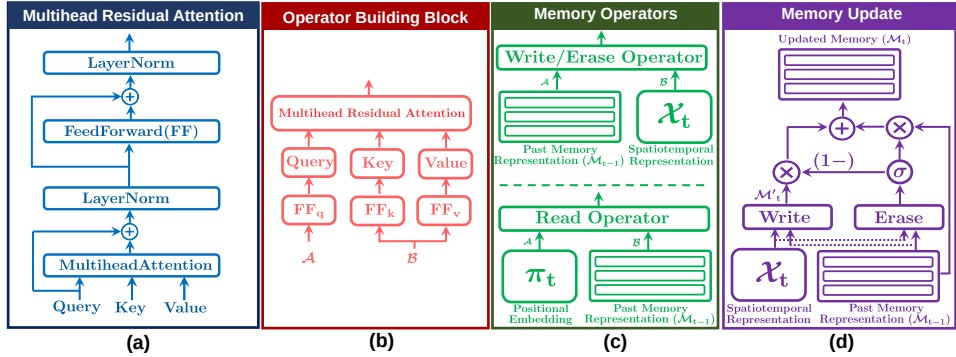

Figure 4: **Details of the memory operations.** **(a)** block diagram of the multihead residual attention, **(b)** composition of an `Operator`$(\mathcal{A}, \mathcal{B})$, **(c)** `Read` operator calculates query from positional embedding $\pi_t$ and key-value pair from past memory representation, $\mathcal{M}_{t-1}$. The `Write` and `Erase` operator computes query from $\mathcal{M}_{t-1}$ and key-value pair from $\mathcal{X}_t$, and, **(d)** An update to the associative memory occurs with a linear combination of the new value to be added $\mathcal{M}'_t$ and past representation $\mathcal{M}_{t-1}$ weighted by an elementwise scaling factor generated by the `Erase` operator.

projecting $Q$, $K$, and $V$ onto $h$ separate $d/h$ dimensional space, respectively. The output is a linear combination of $h$-different attentions (`Attn`$(.,.,.;a)$) applied to each of these $h$ projections:

$$\texttt{MultiHead}(Q, K, V; w, a) = \texttt{concatenate}(O^1, ..., O^h)W^o \tag{2}$$

$$\text{where} \quad O^i = \texttt{Attn}(QW_i^q, KW_i^k, VW_i^v; a_i) \tag{3}$$

$w = \{W_i^q, W_i^k, W_i^v\}_{i=1}^h \in \mathbb{R}^{d \times \frac{d}{h}}$ and $W^o \in \mathbb{R}^{hd_v \times d}$ are learnable parameters. A following residual block completes the operation and generates output, $O$ as follows:

$$O = \texttt{Operator}(\mathcal{A}, \mathcal{B}) := \texttt{LayerNorm}(O_h + \texttt{FF}^o(O_h)) \tag{4}$$

$$\text{where} \quad O_h = \texttt{LayerNorm}(\mathcal{A} + \texttt{MultiHead}(Q, K, V; w, a)) \tag{5}$$

`LayerNorm` is layer normalization function (Ba et al., 2016). Figure 4 illustrates these operations.

***Positional encoder***: EventFormer generates positional embedding by mapping the event sequence from a low-dimensional (2D) space to a higher-dimensional feature space. This is done by a learnable Fourier feature-based positional encoder. At any given time $t$, the positional encoder, $\Pi : x_t \in \mathbb{R}^{n \times 2} \rightarrow \pi_t \in \mathbb{R}^{n \times d}$ parameterized by $W_p \in \mathbb{R}^{\frac{d}{2} \times 2}$ maps a list of $n$ 2-dimensional events, $x_t$ to a $d$-dimensional feature space, $\pi_t$ using the following equation:

$$\pi_t = \Pi(x_t) := \frac{1}{\sqrt{d}} \left( \texttt{concatenate} \left( \cos(x_t W_p^T), \sin(x_t W_p^T) \right) \right) \tag{6}$$

***Spatial correlation:*** To compute a refined representation that considers spatial correlation among the events, we compute self-attention among the positional embedding $\pi_t$ by using the `Refine` operator. The efficient attention mechanism modifies the Equation 1 as follows:

$$\texttt{EfficientAttn}(Q, K, V; a_q, a_k) = a_q(Q) \left( a_k \left( K^\top \right) V \right) \tag{7}$$

where $a_q$ and $a_k$ denote the row-wise and column-wise scaled softmax activation, respectively. A closer look at Equation 7 reveals that the memory and computation complexity has been reduced to $\mathcal{O}(dn + d^2)$ and $\mathcal{O}(d^2 n)$ respectively, which is linear with respect to the number of events, $n$. Finally, we stack multiple `Refine` operators to compute $\mathcal{Z}_t \in \mathbb{R}^{n \times d}$ as their higher-order interactions.

***Associative memory augmented recurrent module:*** The proposed associative memory has a *query-key* association-based memory access architecture with a separate association mechanism for state retrieval and update operation (Figure 4). We define $\mathcal{M}_t \in \mathbb{R}^{m \times d}$ as the stored memory representation at time $t$. The `Read` operator retrieves the past representation $\mathcal{H}_{t-1} \in \mathbb{R}^{n \times d}$ using the current event locations $\pi_t$. It computes the query vectors from $\pi_t$ and key, value vectors using the past stored representation $\mathcal{M}_{t-1}$. The final retrieved state is a weighted sum of the projected $\mathcal{M}_{t-1}$

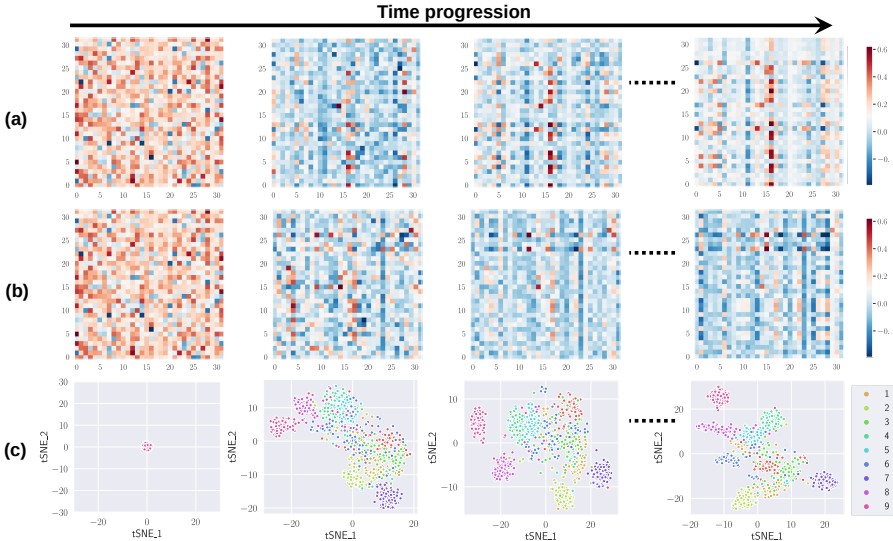

Figure 5: **Temporal evolution of memory representation. (a)** and **(b)** Change in memory representation over time for samples from two different classes. **(c)** tSNE plot of the memory representation for 10 randomly selected classes from N-Caltech101 dataset.

where the weights are computed through the association between the event locations ($\pi_t$) and the abstract memory addresses (keys projected from $\mathcal{M}_{t-1}$). This implies that a particular memory representation gets more weight if its corresponding address (key) has a higher similarity with the query (i.e., positional embedding). The retrieved $\mathcal{H}_{t-1}$ is used as the past hidden state of the recurrent module, $\mathcal{R}$, to compute the current spatiotemporal representation, $\mathcal{X}_t \in \mathbb{R}^{n \times d}$. We use gated recurrent unit (GRU) as the recurrent module. The `Write` and `Erase` operator jointly update $\mathcal{M}_t$ with new information. The `Write` operator computes the query vectors from $\mathcal{M}_{t-1}$ and the key-value pair from $\mathcal{X}_t$ to query the location of the memory that needs to be updated and generate the new representation, $\mathcal{M}'_t \in \mathbb{R}^{m \times d}$. The purpose of `Erase` operator is to calculate a set of element-wise scaling factors, $\alpha_t \in \{\mathbb{R}^{m \times d} \mid 0 \leq \alpha_t \leq 1\}$ to control the amount of update at each memory location. Additional details can be found in Appendix I. Finally, we update the memory using:

$$\mathcal{M}_t = \alpha_t \mathcal{M}_{t-1} + (1 - \alpha_t)\mathcal{M}'_t \tag{8}$$

***Classification head:*** A single feedforward layer $\texttt{FF}_{pred}(.)$ that maps the flattened memory representation, $\mathcal{M}_t^{flat} \in \mathbb{R}^{md}$ to the prediction vector, $Y_t \in \mathbb{R}^c$ where $c$ is the number of classes.

## 3 EXPERIMENTS AND RESULTS

***Datasets and metrics:*** We evaluate the performance of our method on standard event-based object recognition benchmarks: N-Caltech101 and N-Cars (see Appendix D for details). We consider two metrics: recognition accuracy and the number of floating point operations required to update the representation for each new event (MFLOPs/ev).

***Implementation***: EventFormer follows an end-to-end feed-forward layer-based implementation. We use 32×32 and 16×16 as memory dimensions ($\mathbb{R}^{m \times d}$) for N-Caltech101 and N-Cars dataset, respectively. Unless otherwise specified, all the multihead residual attention blocks use $h$ (number of heads) = 4, and the number of stacks for the `Refine` operator, $R = 2$. Additional details including training hyperparameters and time intervals can be found in Appendix E.

***Representation Learning in Associative Memory:*** We first visualize the evolution of the learned representation in the associative memory for different classes of N-Caltech101 (Figure 5). Initially, the memory representation starts from the same initial state for all classes. As events are observed over time, the memory states start to form unique patterns for different classes. The tSNE plot in

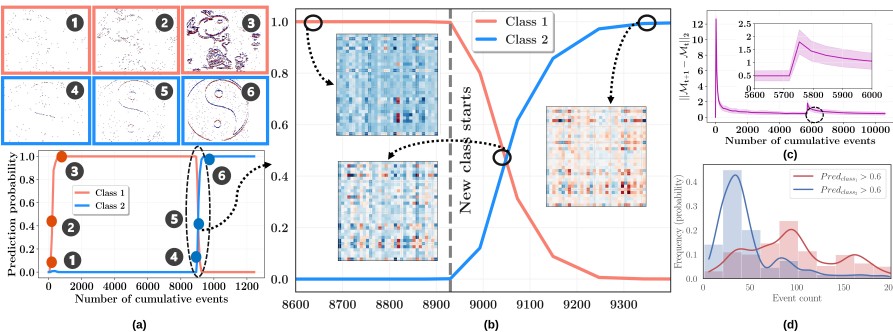

Figure 6: **Memory representation dynamics over sequential class update**. **(a)** Change in prediction probability of the target class over time. **(b)** Change in memory representation when class changes occur. **(c)** Average memory update activity for a large-scale experiment similar to (a) and (b), and **(d)** their corresponding perception latency distribution.

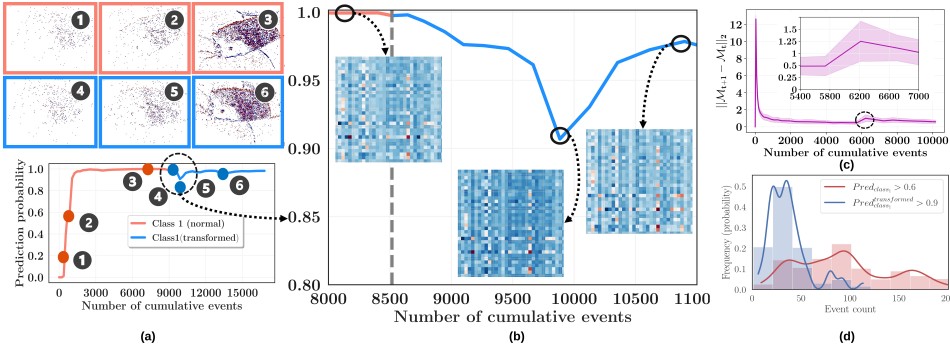

Figure 7: **Memory representation dynamics over sequential input transformation**. **(a)** Change in prediction probability of the target class over time. **(b)** Change in memory representation when transformed input appears. **(c)** Average memory update activity for a large-scale experiment similar to (a) and (b), and **(d)** their corresponding perception latency distribution.

Figure 5(c) shows that the learned representations for various classes form a distinct cluster enabling separation of the class boundaries even with a relatively simple classifier head.

***Temporal Update of the Learned Representation with Class Change:*** We study the temporal updates of the memory states when samples from two different classes are passed to EventFormer in a streaming manner (Figure 6). The memory updates its representation as the object class changes, enabling accurate classification results over time. We randomly take 3000 samples from the N-Caltech101 dataset consisting of sequential class change (3000 unique class combinations). We plot the distribution of the number of events required for the target class output to reach a 0.6 confidence, hereafter referred to as the perception latency (Figure 6 (d)). The median perception latency is $\sim 120$ and $\sim 60$ events for the initial and new class, respectively. The perception latency for the initial class (red) has a wider distribution since the memory has to adapt its representation from the initial state. We also plot the mean and standard deviation of the Frobenius norm of the difference between two successive memory representations $\|\mathcal{M}_{t+1} - \mathcal{M}_t\|_2$, hereafter referred to as the memory update activity, for the same experiment (Figure 6 (c)). Higher memory update activity, observed when developing the representation of the initial class, stabilizes with more events of the same class. The change in the class also results in increased memory update activity but less than the initial case. This shows that EventFormer can adapt to new representations in real-time and re-use important features already computed in the past thereby reducing perception latency.

***Temporal Update of Learned Representation with Input Transformation:*** We study the temporal update of the memory states over time while objects from the same class but with different transformations (rotation) are passed to EventFormer in a streaming manner (Figure 7). With the incoming events from the rotated sample, we observe degradation in prediction probability due to represen-

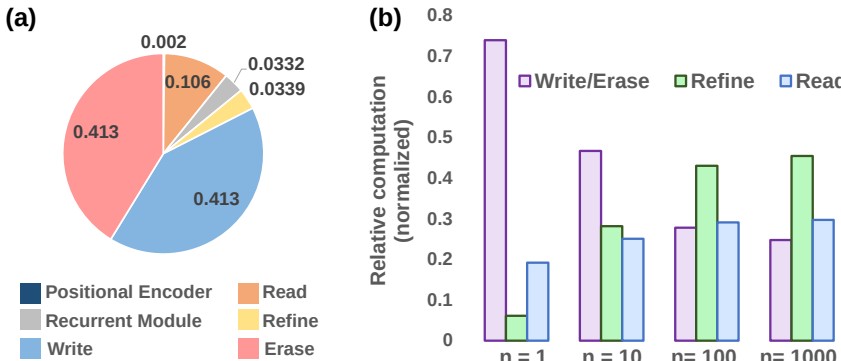

Figure 8: **FLOP analysis.** (a) Normalized FLOP contribution from the key components of Event-Former (for $n$=1). (b) Change in relative compute cost for the `Operators` with increasing $n$.

| Experiment | Associative memory | Separate write and erase operator | Refine operator | Accuracy ↑ | MFLOPs/ev ↓ |
|:---:|:---:|:---:|:---:|:---:|:---:|
| (a) | ✗ | ✗ | ✓ | 0.670 | 0.014 |
| (b) | ✓ | ✗ | ✓ | 0.752 | 0.041 |
| (c) | ✓ | ✓ | ✗ | 0.703 | 0.035 |
| **(d)** | ✓ | ✓ | ✓ | **0.848** | **0.048** |

Table 1: Ablation study of various key components of EventFormer on the N-Caltech101 dataset.

tation update, which stabilizes over time. We consider 3000 random samples from N-Caltech101 with combinations of two random rotations between 10 to 30 degrees. We observe that the memory update activity is less for input transformation of the same example (Figure 7 (c)), compared to a completely new example (Figure 6 (c)). This shows that EventFormer can preserve learned representation for a class even under input transformation enabling less memory update activity.

***Computational Complexity***: We analytically derive the computational complexity model of Event-Former (details in Appendix C). We study the FLOP contributions from the key components in Figure 8. For a single event ($n$=1), most of the FLOP costs are from the memory update block due to its $\mathcal{O}(md^2 + md + d^2)$ compute complexity. The cost of memory update diminishes with a larger $n$ while the `Read` and `Refine` operator starts to dominate (Figure 8 (b)).

***Ablation study on the components:*** We consider the experiments where EventFormer is implemented without *(a):* the associative memory, *(b):* separate `Write` and `Erase` operator, and *(c):* `Refine` operator and compare the results with *(d):* the complete EventFormer to analyze the impact of different components. For experiment (a), we pool the $\mathcal{X}_t$ to a 1-dimensional vector (Vemprala et al., 2021) to deal with the variable-sized sequence. Also, we make the hidden size of the recurrent module $m \times d$ so that the `FF_pred` receives the same-sized input. In experiment (b), we linearly project the output of the `Write` operator into a $\mathbb{R}^{m \times 2d}$

| Hyperparameter | | Accuracy ↑ | MFLOPs/ev ↓ |
|:---:|:---:|:---:|:---:|
| | 1 | 0.759 | 0.042 |
| $R$ | 2 | 0.848 | 0.048 |
| | 4 | 0.864 | 0.061 |
| | 16 | 0.550 | 0.012 |
| $m(=d)$ | 32 | 0.848 | 0.048 |
| | 48 | 0.882 | 0.108 |

Table 2: Performance of EventFormer on N-Caltech101 with different hyperparameters.

space and split it into two equal-sized vectors to get $\mathcal{M}'_t$ and $\alpha_t$. The results are shown in Table 1. Merging the write and erase operation marginally reduces the MFLOPs/ev, but at the expense of less flexibility during memory updates and hence, lower accuracy. Without the `Refine` operator, the model saves more computation but lacks the higher-order interaction modeling among the events, causing marginal accuracy loss. Finally, absence of associative memory reduces maximum compu-

| Methods | Representation | Async. | N-Caltech101 | | N-Cars | |
|---|---|---|---|---|---|---|
| | | | Accuracy ↑ | MFLOPs/ev ↓ | Accuracy ↑ | MFLOPs/ev ↓ |
| H-First | Spike | ✓ | 0.054 | - | 0.561 | - |
| Gabor-SNN | Spike | ✓ | 0.284 | - | 0.789 | - |
| HOTS | Time-Surface | ✓ | 0.210 | 54.0 | 0.624 | 14.0 |
| HATS | Time-Surface | ✓ | 0.642 | 4.3 | 0.902 | 0.03 |
| DART | Time-Surface | ✓ | 0.664 | - | - | - |
| EST | Event-Histogram | ✗ | 0.817 | 4150 | 0.925 | 1050 |
| Matrix-LSTM | Event-Histogram | ✗ | 0.843 | 1580 | 0.926 | 1250 |
| YOLE | Voxel-Grid | ✓ | 0.702 | 3659 | 0.927 | 328.16 |
| AsyNet | Voxel-Grid | ✓ | 0.745 | 202 | 0.944 | 21.5 |
| EvS-S | Graph | ✓ | 0.761 | 11.5 | 0.931 | 6.1 |
| AEGNN | Graph | ✓ | 0.668 | 0.369 | **0.945** | 0.03 |
| **Ours** | Unstructured Set | ✓ | **0.848** | **0.048** | 0.943 | **0.013** |

Table 3: Performance comparison with state-of-the-art event-based and dense methods.

tation but also causes the model to lose spatial information (due to the pooling operation) resulting in minimum performance.

***Effect of Hyperparameters:*** We study the impact of hyperparameters $R$ and $m(= d)$ (Table 2). We observe that increasing their values result in better performance but at the cost of higher MFLOPs/ev. Therefore, we follow our initial setting ($R$=2 and $m(= d)$=32) to have a better balance between accuracy and compute cost while comparing with the state-of-the-art methods.

***Comparison with the state-of-the-art (SoTA):*** We compare EventFormer with SoTA dense and event-based methods on these two datasets (Table 3). Following the previous work (Schaefer et al., 2022), we report the average MFLOPs/ev on a window of 25000 events. Methods utilizing hand-crafted features: H-First (Orchard et al., 2015), Gabor-SNN (Bovik et al., 1990), HOTS (Lagorce et al., 2016), HATS (Sironi et al., 2018), DART (Ramesh et al., 2019), require fewer MFLOPs/ev while performing worse than the data-driven methods. EST (Gehrig et al., 2019) and Matrix-LSTM (Cannici et al., 2020b) are synchronous methods that learn the optimal representation and achieve higher accuracy. However, they require compute-intensive feature extractors to work on the learned representations. Whereas EventFormer does not require any complex feature extractors since the learned representation in the memory captures both the spatial and temporal information. It can achieve similar performance with $30000\times$ less computation. YOLE (Cannici et al., 2019) and AsyNet (Messikommer et al., 2020) modify the CNNs to enable asynchronous, sparse processing to reduce the compute cost. However, they do not consider the temporal information in the events. EvS-S (Li et al., 2021c) and AEGNN(Schaefer et al., 2022) are graph-based methods with asynchronous and efficient graph nodes update. However, they use sub-sampling to restrict the graph size from growing prohibitively large, leading to suboptimal performance. Our method, on the other hand, inherently learns to leverage useful events to update the memory without requiring any redundant computation (such as radius search). The memory mechanism enables learning better temporal correlation compared to the graph-based methods. As a result, our method achieves $9\%$ better accuracy on N-Caltech101 while being $200\times$ more efficient compared to (Li et al., 2021c).

***Computation Latency:*** The latency of EventFormer implemented with PyTorch on an Nvidia RTX3090 is 4.5ms/event on N-Caltech101 dataset which is $10\times$ faster than the existing most efficient method (Schaefer et al., 2022)(52ms/event). Unlike their method, we do not require searching over a large graph to update the representation which reduces computational latency.

## 4 CONCLUSION AND FUTURE WORK

We propose a novel memory-augmented representation learning framework–EventFormer, for asynchronous and efficient event-based perception. EventFormer learns to store, retrieve and update its memory representation in the latent form of higher-order spatiotemporal dynamics of the events that allow it to achieve high performance with minimal compute cost. Future works include EventFormer on more complex tasks including object detection, depth estimation, and optical flow prediction.

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

# Supplement to "Associative Memory Augmented Asynchronous Spatiotemporal Representation Learning for Event-based Perception"

## A  APPENDIX

In this supplementary material, we provide experimental and additional details on EventFormer.

## B  DETAILS OF THE NOTATION USED IN THIS WORK

| Type | Description | Notation | Dimension |
|------|-------------|----------|-----------|
| Scaler parameters | number of events | $n$ | - |
| | dimension of event-representation | $d$ | - |
| | row size of the memory | $m$ | - |
| | number of classes | $c$ | - |
| Functions | positional encoder | $\Pi(.)$ | - |
| | recurrent module | $\mathcal{R}(.,.)$ | - |
| | classifier head | $\text{FF}_{\text{pred}}(.)$ | - |
| Vector/Matrix | sequence of events at time $t$ | $x_t$ | $n \times 2$ |
| | positional embedding | $\pi_t$ | $n \times d$ |
| | associative memory representation at $t$ | $\mathcal{M}_t$ | $m \times d$ |
| | retrieved state from the memory (past hidden state for the recurrent module) | $\mathcal{H}_{t-1}$ | $n \times d$ |
| | representation with higher order interaction among the events | $\mathcal{Z}_t$ | $n \times d$ |
| | current spatiotemporal representation | $\mathcal{X}_t$ | $n \times d$ |
| | new memory representation at $t$ | $\mathcal{M}'_t$ | $m \times d$ |
| | scaling factors to control memory update | $\alpha_t$ | $m \times d$ |
| | prediction at time $t$ | $Y_t$ | $1 \times c$ |

Table 4: Different notations used to describe the operations of EventFormer

## C  DERIVATION OF COMPUTATIONAL COMPLEXITY MODEL OF EVENTFORMER

Here we derive the computational complexity model of the proposed EventFormer architecture. Let us define $n$ as the number of events to be processed within time-interval $\tau$, $d$ as the dimension of the representation, $m$ as the number of rows in the memory, $h$ be the number of heads in the residual attention blocks (we use the same $h$ in all the blocks), $r$ as the number of stacks for the Refine operator, and $c$ be the number of class for the recognition task.

**(i) Positional Encoder:** Our positional encoder block consists of single matrix multiplication with the input, that is to compute $x_t W_p^T$, we need $2nd$ multiplications and $nd(2-1)$ additions, so in total $3nd$ FLOPs.

**(ii) Read Operator:** Read operator consists of a residual multihead attention block with $h$ heads. To begin with, calculating the query matrix for each head, we need $n(\frac{d}{h})^2$ multiplication and $n\frac{d}{h}(\frac{d}{h}-1)$ additions, so in total $n\frac{d}{h}(2\frac{d}{h}-1)$ FLOPs. Similarly for the key and value matrix, we need total $2m\frac{d}{h}(2\frac{d}{h}-1)$ FLOPs. Then, to compute $QK^\top$, we need an additional $mn(2\frac{d}{h}-1)$ computation. Similarly the product $(QK^\top)V$ requires $n\frac{d}{h}(2m-1)$ FLOPs. Upto now for each head, we require total FLOPs:

$$n\frac{d}{h}(2\frac{d}{h} - 1) + 2m\frac{d}{h}(2\frac{d}{h} - 1) + mn(2\frac{d}{h} - 1) + n\frac{d}{h}(2m - 1),$$

Therefore, for $h$ heads, it becomes:

$$nd(2\frac{d}{h} - 1) + 2md(2\frac{d}{h} - 1) + hmn(2\frac{d}{h} - 1) + nd(2m - 1)$$

The following 2 residual connection involves pointwise addition of a $n \times d$ dimensional matrix and the out-projection involves an additional $nd(2d - 1)$ FLOPs. Finally, total FLOPs count for the Read($\pi_t, \mathcal{M}_{t-1}$) becomes:

$$nd(2\frac{d}{h} - 1) + 2md(2\frac{d}{h} - 1) + hmn(2\frac{d}{h} - 1) + nd(2m - 1) + 2nd + 2nd(2d - 1)$$

**(iii) Recurrent Module:** Our recurrent module is a GRU that involves 3 matrix multiplication (update, reset, hidden) for the current input, $\pi_t$ and hidden state $H_{t-1}$ each. Therefore 6 matrix multiplication in total followed by 5 elementwise addition and 2 elementwise multiplication. Therefore, the total computation becomes:

$$6nd(2d - 1) + 7nd$$

**(iv) Refine Operator:** Unlike the Read($.,.$) operator, it projects the same input, $X_t$ to query, key and value space. Therefore, total computation for such project becomes $3n\frac{d}{h}(2\frac{d}{h} - 1)$, followed by additional $(\frac{d}{h})^2(2n - 1)$ and $n\frac{d}{h}(2\frac{d}{h} - 1)$ FLOPs for $K^\top V$, and $Q(K^\top V)$ computation. Therefore, for $h$ heads and considering the additional cost for residual operations and out-projections, the final cost becomes:

$$r\left(3nd(2\frac{d}{h} - 1) + (\frac{d^2}{h})(2n - 1) + nd(2\frac{d}{h} - 1) + 2nd + 2nd(2d - 1)\right)$$

here $r$ accounts for the number of stacks.

**(iv) Write and Erase Operator:** Similar to the Read($.,.$) operator, we can calculate the total FLOPs required for the Write($\mathcal{M}_{t-1}, Z_t$) and Erase($\mathcal{M}_{t-1}, Z_t$) operation combined as follows:

$$2\left(md(2\frac{d}{h} - 1) + 2nd(2\frac{d}{h} - 1) + hmn(2\frac{d}{h} - 1) + md(2n - 1) + 2md + 2md(2d - 1)\right) + 3md$$

here, the additional $3md$ accounts for the 2 pointwise addition and 1 multiplication in Equation 8.

**(iv) Perception Output:** The final output is calculated by a vector($\in \mathbb{R}^{md}$)-matrix($\in \mathbb{R}^{md \times c}$) multiplication, which requires an additional $c(2md - 1)$ FLOPs.

**Total FLOPs:** The total FLOPs accounting all the above computation becomes:

$$\underbrace{3nd}_{\text{Positional Encoder}} + \underbrace{nd(2\frac{d}{h} - 1) + 2md(2\frac{d}{h} - 1) + hmn(2\frac{d}{h} - 1) + nd(2m - 1) + 2nd + 2nd(2d - 1) +}_{\text{Read Operation}}$$

$$\underbrace{6nd(2d - 1) + 7nd}_{\text{Recurrent Module}} + \underbrace{r\left(3nd(2\frac{d}{h} - 1) + (\frac{d^2}{h})(2n - 1) + nd(2\frac{d}{h} - 1) + 2nd + 2nd(2d - 1)\right)}_{\text{Refine Operator}}$$

$$+ \underbrace{2\left(md(2\frac{d}{h} - 1) + 2nd(2\frac{d}{h} - 1) + hmn(2\frac{d}{h} - 1) + md(2n - 1) + 2md + 2md(2d - 1)\right) + 3md}_{\text{Memory Update (includes both Write and Erase operation)}}$$

$$+ \underbrace{c(2md - 1)}_{\text{Output}}$$

$$(9)$$

## D DETAILS OF THE DATASETS

*N-Caltech101* (Orchard et al., 2015) is converted to event space from standard frame-based dataset (Caltech101). It has 8246 event sequences from 101 class categories following the original Caltech101 dataset. N-Caltech101 uses an event camera to record event streams generated by moving images displayed on a monitor.

*N-Cars* (Sironi et al., 2018) is a real-world event-camera dataset with 24039 event sequences from 2-class instances (car or background). N-Cars utilizes an event camera mounted on a moving car recording its surroundings in a real-world setting.

## E TRAINING DETAILS

*Data Preparation:* For both of the datasets, we use the official test set portion to report the result and split the remaining into 90% and 10% ratios for training and validation purposes. The event coordinates are normalized by dividing them by the height and width ($180 \times 240$ for N-Caltech101 and $100 \times 120$ for N-Cars) of the frame resolution. We use a 50 ms long sequence for each sample to reduce training complexity. Each sample is chunked by a 1ms window, resulting in 50-sequences for each sample.

*Optimization:* We use the standard categorical cross-entropy as the loss function to train the network with Adam (Kingma & Ba, 2014) with batch size 128 for N-Cars and 64 for N-Caltech101 with an initial learning rate of $1e^{-3}$ that decreases by a factor of 5 after every 25 epochs. The memory representation is initialized by a set of learnable parameters $\mathcal{M}_I \in \mathbb{R}^{m \times d}$. We also use a dropout of 0.2 on the memory representation before passing them to the final classification layer. To avoid gradient explosion during training, we use gradient clipping with a maximum gradient norm of 100.

## F ADDITIONAL ABLATION EXPERIMENT

**Impact of polarity**: In our current formulation, we did not consider polarity information of the events during the event-encoding part. We conduct an additional experiment to evaluate the impact of polarity by taking them as an additional input alongside the positional coordinates of the events. In this formulation, we consider $\mathcal{E}_\tau = \{(x_i, y_i, p_i)\}$ where $p_i$ denotes the polarity of the event at the $(x_i, y_i)$ location. At a given time $t$, while we use the same positional encoder, $\Pi$ to process $x_t$, we use a separate MLP layer parameterized by $W_{pol} \in \mathbb{R}^{d \times 1}$ to map the list of $n$ event polarities, $p_t$ to a $d$-dimensional feature space, $po_t$. Finally, we add $\pi_t$ and $po_t$ together to generate a refined $\pi_t$ that now contains both the polarity and positional information. We keep the subsequent operations unchanged. Following the same training and evaluation procedure on N-Caltech101 dataset, this modified architecture achieved a slightly better accuracy of $0.849$ (compared to the original $0.848$) with negligible additional compute cost.

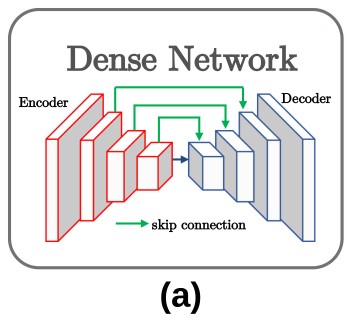
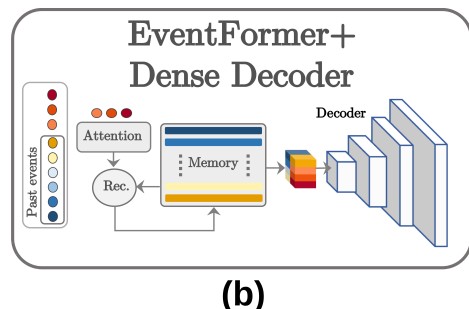

**(a)**        **(b)**

Figure 9: **Network architectures for dense prediction tasks. (a)** Convolution-based hierarchical encoder-decoder structure with skip connections in between. **(b)** Possible adaptation of EventFormer to work with existing decoder structure with no skip-connection.

## G    FEASIBILITY STUDY OF APPLYING EVENTFORMER ON COMPLEX TASK

**Motion Segmentation:** Our primary focus in this work was to develop an event-based spatiotemporal representation learning framework and we used classification as an example task to demonstrate the efficacy of our method. In this section, we further evaluate our method on a dense task–motion segmentation as an example of tasks more complex than classification. Unlike the classification task, motion segmentation is a dense prediction task. Existing works (Mitrokhin et al., 2019; Sanket et al., 2020; Parameshwara et al., 2021) on such dense task leverage hierarchical encoder-decoder network architecture where the encoder encodes the high-dimensional space into a compact latent-space while the decoder decodes it back to its original shape. Therefore, applying Event-Former on such dense tasks requires its own decoder to reconstruct the dense output from its latent memory representation.

| Method | IoU | | MFLOPs/ev |
| --- | --- | --- | --- |
| | boxes | floor | (Encoder) |
| EV-IMO | 0.70 | 0.59 | 4786.7 |
| EVDodgeNet | 0.67 | 0.61 | 2575.0 |
| SpikeMS | 0.59 | 0.46 | 18.5 |
| EventFormer (m=64,d=32) | 0.47 | 0.44 | 0.064 |
| EventFormer (m=64,d=64) | 0.49 | 0.43 | 0.193 |
| EventFormer (m=256,d=32) | 0.55 | 0.53 | 0.160 |

Table 5: Comparison of preliminary Event-Former adaptation with other methods on EV-IMO dataset.

While designing a novel decoder architecture is out of the scope of this work, we use the existing convolution-based decoder structure for this experiment. We reshape the memory representation $\mathcal{M} \in \mathbb{R}^{m \times d}$ into $\mathcal{M}_n \in \mathbb{R}^{a \times b \times d}$ (where $m = a \times b$) to give it a 2D-positional bias and pass it to the decoder as shown in Figure 9 so that end-to-end training with the decoder enables it to learn the required 2D-latent representation in the encoded space.

**Dataset and Experimental Setup:** We train and evaluate our method on a sub-set of EV-IMO dataset (Mitrokhin et al., 2019). EV-IMO consists of challenging scenarios where different objects move at varying speeds and directions. A monocular event-based camera captures the motions. Since our primary objective for this experiment is only to provide some insights on how to scale our method on complex tasks, we use a subset of the available five different sequences (boxes and floor) to reduce the training time. We center-crop the events by a $256 \times 256$ spatial window (i.e., we do not consider the events that fall outside this window) and use a $20ms$ long sequence for every sample. Each sequence is chunked by a $1ms$ temporal window resulting in 20 sequences per sample. We use Adam with a learning rate of $5 \times 10^{-3}$ and trained for 80 epochs. We consider the Intersection over Union (IoU) and MFLOPs/ev as the performance metrics. Since all the methods (including ours) employ similar decoder structures, we consider only the encoder part to compare the MFLOPs/ev.

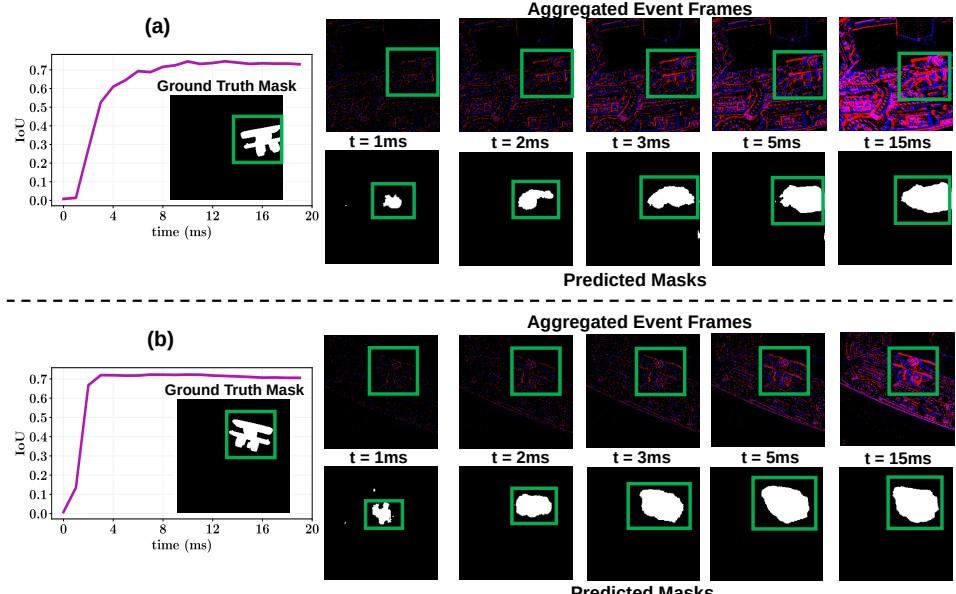

Figure 10: **Qualitative motion-segmentation performance on EV-IMO dataset.**

**Preliminary Performance:** We compare the performance with existing state-of-the-art dense and event-based motion-segmentation methods (Table. 5). Both the EV-IMO (Mitrokhin et al., 2019) and EVDodgeNet (Sanket et al., 2020) are dense processing-based methods that aggregate events into a frame-based representation. SpikeMS (Parameshwara et al., 2021) considers the incoming events as spikes and adopts a spiking-neural-network (SNN) based encoder-decoder structure to process them asynchronously. For our method, we consider three different settings: $m = 64, a = b = 8, d = 32$, $m = 64, a = b = 8, d = 64$, and $m = 256, a = b = 16, d = 32$. We observe that the better performance of the dense methods comes with the cost of much higher computational complexities. Our method can

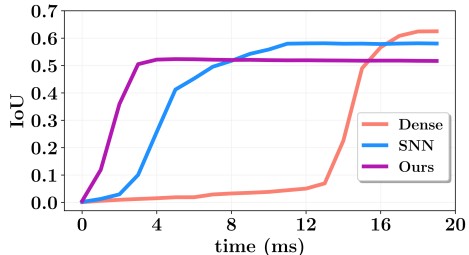

Figure 11: **Incremental prediction performance comparison.** Our method can converge to its peak performance significantly faster than the existing methods.

achieve comparable performance with SpikeMS while being $115\times$ more efficient. We also observe that the performance of our method improves with higher memory capacity across the space. Another interesting observation is that the performance gain from higher $m$ is much larger compared to the higher $d$. This implies that representation with more fine-grained spatial information is necessary for dense prediction tasks. This is also helpful in terms of computational benefits since our FLOP model has a sub-linear relationship with $m$. Our unstructured formulation enables us to perform incremental prediction, even for dense tasks. Figure. 11 shows the comparison with existing methods on such incremental prediction tasks. For the dense method, we integrate the events with incremental time-window starting from $1ms$ up to $20ms$ with $1ms$ interval. As shown, our method can outperform both methods at the early stage, requiring much fewer events. We also visualize the qualitative performance of our method in Figure. 10. Although our method can achieve high IoU very fast, it fails to capture the local details of the object. This is because, unlike existing methods, our decoder can not utilize any skip connections. Such skip connections are necessary for the existing encoder-decoder network architectures to recover the local spatial information that may have been lost during the encoding process (Drozdzal et al., 2016). These results indicate that existing decoder architectures may not be optimum to work on our representation and novel architectural innovations are required in this regard. We leave this as a potential future research direction.

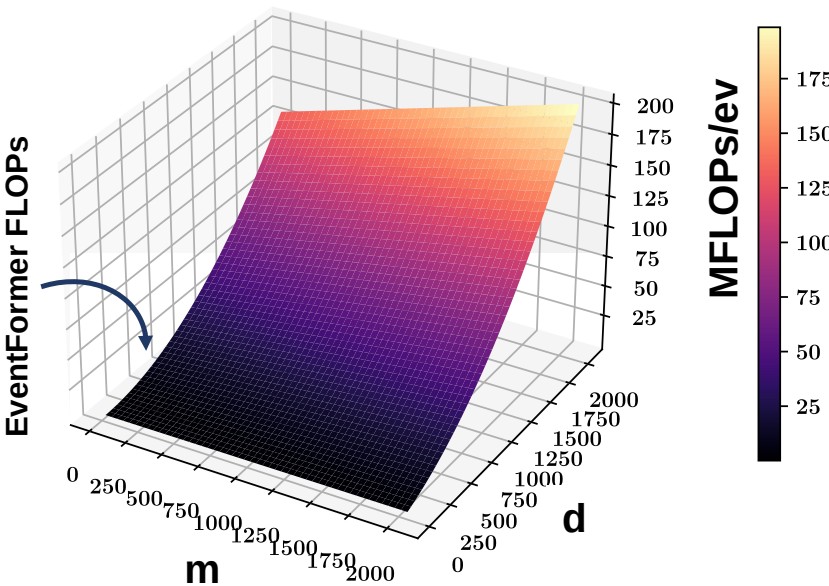

Figure 12: **Scaling of EventFormer compute cost with increasing $m$ and $d$ from 16 to 2048.**

**Scaling of Computational Complexity:** Our compute model derived in Appendix C shows exponential relation with $d$ while sub-linear relation with $m$. For more complex and dense tasks with higher spatial resolution, EventFormer may need to scale up its memory size at the cost of additional compute overhead. We conduct an additional experiment to better understand how our computational cost would increase with higher memory size, and the result is shown in Figure 12. We observe that the compute cost increases at a significantly lower rate for higher $m$ compared to a higher $d$. For both $m = d = 2048$, our EventFormer formulation requires about 210 MFLOPs/ev, which is still $10\times$ lower compared to the existing dense encoder architecture of EveDodgeNet (Sanket et al., 2020).

## H    LATENCY SCALING FOR LARGER EVENT COUNTS

We perform additional timing experiments to understand the change in latency with an increasing number of events. Table 6 shows the result. We can observe that our latency has a highly sub-linear relationship with respect to the number of events. Also, the throughput increases with the number of events thanks to the parallel processing capabilities of the modern hardware accelerators (GPU in our case). We also measure the time required for a dense method (Matrix-LSTM (Cannici et al., 2020b)) to process a block of 30000 events which is $35.93ms$ for N-Caltech101 dataset. It is noteworthy to mention that Matrix-LSTM utilizes its own highly optimized CUDA kernels to reduce their latency significantly. While developing such an optimized kernel for EventFormer is out of the scope of this work, we also expect a similar latency reduction for our method through such software optimization.

| n | Latency (ms) | Throughput (Kev/s) |
|---|---|---|
| 1 | 4.53 | 0.22 |
| 10 | 4.62 | 2.16 |
| 100 | 4.78 | 20.92 |
| 1000 | 4.91 | 203.66 |
| 10000 | 7.54 | 1333.33 |
| 20000 | 14.15 | 1413.42 |
| 30000 | 20.8 | 1442.31 |

Table 6: EventFormer latency and throughput with an increasing number of events.

| Methods | Representation | Async. | N-Caltech101 | | N-Cars | |
|---|---|---|---|---|---|---|
| | | | Accuracy ↑ | Latency(ms) ↓ | Accuracy ↑ | Latency(ms) ↓ |
| H-First | Spike | ✓ | 0.054 | - | 0.561 | - |
| Gabor-SNN | Spike | ✓ | 0.284 | - | 0.789 | 0.071 (Intel i7) |
| HOTS | Time-Surface | ✓ | 0.210 | - | 0.624 | 0.038 (Intel i7) |
| HATS | Time-Surface | ✓ | 0.642 | - | 0.902 | 0.002 (Intel i7) |
| DART | Time-Surface | ✓ | 0.664 | - | - | - |
| EST | Event-Histogram | ✗ | 0.817 | - | 0.925 | 0.001(RTX2080Ti) |
| Matrix-LSTM | Event-Histogram | ✗ | 0.843 | - | 0.926 | 0.002(GTX1080Ti) |
| YOLE | Voxel-Grid | ✓ | 0.702 | - | 0.927 | - |
| AsyNet | Voxel-Grid | ✓ | 0.745 | - | 0.944 | - |
| EvS-S | Graph | ✓ | 0.761 | 0.004(Intel i7) | 0.931 | - |
| AEGNN | Graph | ✓ | 0.668 | - | **0.945** | - |
| **Ours** | Unstructured Set | ✓ | **0.848** | **0.0007 (RTX3090)** | 0.943 | **0.0005 (RTX3090)** |

Table 7: Latency and Accuracy comparison with state-of-the-art event-based and dense methods.

# I   DETAILS OF THE MEMORY OPERATORS:

**Read:** During Read operation, we want to know: *what are the past states at the current event location?* To do so, we use the multihead residual attention block to perform query-key-based associations in the memory. To be more specific, we query the past memory representation $\mathcal{M}_{t-1}$ using the positional embedding of the current event locations, $\pi_t$. The complete retrieval of the past hidden representation, $\mathcal{H}_{t-1}$ involves the following operations:

$$H_{t-1} = \texttt{Read}(\pi_t, \mathcal{M}_{t-1}) := \texttt{LayerNorm}(O_r + \texttt{FF}_r^o(O_r)) \tag{10}$$

$$\text{where} \quad O_r = \texttt{LayerNorm}(\pi_t + \texttt{MultiHead}(Q_r, K_r, V_r; w, a)) \tag{11}$$

Here, $Q_r$ represents the query vector calculated from $\pi_t$, and $K_r, V_r$ represents the key and value vectors computed from the $\mathcal{M}_{t-1}$.

**Write and Erase:** Similar to the Read operator, we adopt residual multi-head attention block for Write operator to calculate the new memory representation, $\mathcal{M}'_t$. However, this time we compute the query vectors from $\mathcal{M}_{t-1}$ and *key-value* pair from the refined spatiotemporal representation, $\mathcal{X}_t$. The idea here is that we want to query the location of the memory that needs to be updated while the contents to be updated are provided by the new representation.

$$\mathcal{M}'_t = \texttt{Write}(\mathcal{M}_{t-1}, \mathcal{X}_t) := \texttt{LayerNorm}(O_w + \texttt{FF}_w^o(O_w)) \tag{12}$$

$$\text{where} \quad O_w = \texttt{LayerNorm}(\mathcal{M}_{t-1} + \texttt{MultiHead}(Q_w, K_w, V_w; w, a)) \tag{13}$$

Here, $Q_w$ represents the query vector calculated from $\mathcal{M}_{t-1}$, and $K_w, V_w$ represents the key and value vectors computed from the $\mathcal{X}_t$. We also introduce Erase operator (follows the same operations of Write operator) that calculates a set of element-wise scaling factors, $\alpha_t \in \{\mathbb{R}^{m \times d} \mid 0 \leq \alpha_t \leq 1\}$ to control the strength of update:

$$\alpha_t = \texttt{sigmoid}(\texttt{Erase}(\mathcal{M}_{t-1}, \mathcal{X}_t)) \tag{14}$$

# J   LATENCY COMPARISON:

A direct comparison of the latency of our and other methods in Table 7 is challenging as different methods have provided their latency measurements using different hardware configurations and software setups. However, we still show the latency/event reported in the original papers for the methods and corresponding hardware configuration. A common methodology was used in all these works where the average latency/event was measured considering the average run time per sample divided by the average number of events per sample.

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
