# OpenReview forum: "Associative Memory Augmented Asynchronous Spatiotemporal Representation Learning for Event-based Perception"
_ICLR.cc/2023/Conference — ICLR 2023 notable top 25%_

### Official Review · Reviewer_rsty · 2022-10-24

**Confidence:** 5
**Correctness:** 3
**Technical Novelty And Significance:** 3
**Empirical Novelty And Significance:** 3
**Recommendation:** 6

**Clarity, Quality, Novelty And Reproducibility:**

The method is mostly quite clear, although some improvements in the read/write/erase section would be appreciated. Overall, the method is compelling, and shows improvements over prior state of the art.

**Strength And Weaknesses:**

The motivation of having a stored memory for event processing is compelling, and the proposed work uses many of the current techniques that have demonstrated efficacy (attention, RNNS). The provided qualitative examples are quite interesting, showing that the memory differentiates between different classes for classification from t-SNE, and can update its state when the input changes to a different class. The results from the classification experiments are also strong, showing strong classification accuracy mostly above prior state of the art, at significantly less compute.

The method is presented as asynchronous, but does seem to require some level of batching of events at input. The Refine stage requires pairwise interactions between the input events, so it seems like this stage would have no effect if a single event was passed in. The computational latency of this method also seems quite high (4.5ms per event), as a single 30k event block would take over two seconds. This seems to go against the arguments presented in Table 3, where efficiency is reported in terms of MFLOPs/ev. For 'synchronous' methods which batch events into tensor representations, they should be able to process 30k event blocks at > 10Hz. Obviously there are tradeoffs for each, but currently the synchronous methods are only described as being more expensive.

Another concern is whether this method would scale up for more difficult tasks. It seems like the memory tensor is at most 32 x 32, which makes sense for classification. However, if a dense task was provided (e.g. optical flow), or something like detection, would it be sufficient to scale up the size of the memory? At what point does this become prohibitively expensive? 32 x 32 seems fairly small for a learned representation.

The outline of the method mostly is clear, but the Read, Write and Erase sections are a little hard to follow. While the other parts are clearly outlined in equations, these three are condensed entirely in text under 'Associative memory augmented recurrent module', and it's not clear what goes where. Some clarifications here could help the reader.

Table 3: AEGNN slightly outperforms the proposed method for N-Cars and should be bolded.

**Summary Of The Paper:**

This paper presents a transformer memory architecture for processing events from an event camera. A block of events are converted into positional embeddings, passed through self attention, and then fused with a memory tensor which is updated with events over time. The memory tensor can then be decoded at any time to produce the desired output. Qualitative examples are provided for the dynamics of the learned memory, and experiments are performed on image classification on the N-Caltech 101 and N-Cars datasets, where the proposed method outperforms previous state of the art, at significantly lower MFLOPs/ev.

**Summary Of The Review:**

I believe that this work can serve as a good model for future memory-based event processing architectures. The overall architecture is simple and well motivated, and the experiments show improvements over prior work. Overall, my main concerns are the claims of efficiency given the latency results and the ability to scale up the method.

---

> ### Author Response · Authors · 2022-11-18
> **Response to the Reviewer rsty 1/2**
>
> We are grateful to the reviewer for their positive feedback about our work. Below we try to address each of your comments:
>
> >The computational latency of this method also seems quite high (4.5ms per event), as a single 30k event block would take over two seconds. This seems to go against the arguments presented in Table 3, where efficiency is reported in terms of MFLOPs/ev. For 'synchronous' methods which batch events into tensor representations, they should be able to process 30k event blocks at > 10Hz. Obviously, there are tradeoffs for each, but currently, the synchronous methods are only described as being more expensive.
>
> We thank the reviewer for this concern. The latency measurement (4.5ms) reported corresponds to the inference time for a single event input to make a fair comparison with AEGNN. We perform additional timing experiments (details in Appendix H) to understand how our latency would change with an increasing number of events. The result is shown below:
>
> | n     | Latency (ms) | Throughput (Kev/s) |
> |----------|--------------|-----------------------|
> | 1         | 4.53         | 0.22                   |
> | 10       | 4.62         | 2.16                   |
> | 100     | 4.78         | 20.92                 |
> | 1000   | 4.91         | 203.66               |
> | 10000 | 7.54         | 1333.33             |
> | 20000 | 14.15        | 1413.42            |
> | 30000 | 20.80        | 1442.31            |
>
> As we can see, the latency increases sub-linearly with respect to the number of events. Overall, we observe that our method is capable of processing 30k event blocks in 20.8ms (at about 50Hz). In contrast, we observe that the ‘synchronous’ methods can process 30k event blocks at >10Hz (27.83Hz or 35.93ms in our measurement). It is important to note that, the considered synchronous method uses highly optimized CUDA kernels to significantly reduce the latency (25%) via software engineering techniques [1]. On the other hand, our method was implemented in pure python PyTorch (and can still perform faster) without any kernel-level or graph-level optimizations. We consider further latency reduction via software engineering (including custom CUDA kernel) as a future development step for this work.
>
> [1] Cannici, Marco, et al. "A differentiable recurrent surface for asynchronous event-based data." ECCV, 2020.

---

> > ### Author Response · Authors · 2022-11-18
> > **Response to the Reviewer 2/2**
> >
> > >Another concern is whether this method would scale up for more difficult tasks. It seems like the memory tensor is at most 32 x 32, which makes sense for classification. However, if a dense task was provided (e.g. optical flow), or something like detection, would it be sufficient to scale up the size of the memory?
> >
> > Reply: We thank the reviewer for this concern. The primary focus of this work was to develop an event-based representation learning framework and study the effectiveness of our representation on classification tasks. We considered extension to more complex tasks as a future research. However, based on all reviewers’ advice, we have performed preliminary experiments to study the feasibility of applying our method to more complex tasks that generate dense outputs. In particular, we developed a model architecture using our representation for motion segmentation using the EV-IMO dataset. However, due to the limitation of time, we experimented with only a subset of the EV-IMO dataset. The preliminary results (detailed in Appendix G) are shown below:
> >
> > |          Method                              |  IoU  |       | MFLOPs/ev (Encoder)|
> > |:------------------------------------:|:---------:|---------|:-----------------------------:|
> > |                                              | boxes | floor   |                                     |
> > | EV-IMO                                | 0.70    | 0.59   | 4786.7                         |
> > | EVDodgeNet                        | 0.67    | 0.61   | 2575.0                         |
> > | SpikeMS                               | 0.59    | 0.46   | 18.5                             |
> > |  EventFormer (m=64,d=32)  | 0.47    | 0.44   | 0.064                           |
> > |  EventFormer (m=64,d=64)  | 0.49    | 0.43   | 0.193                           |
> > | EventFormer (m=256,d=32) | 0.55    | 0.53   | 0.160                           |
> >
> > We observe that our event-based representation can be extended for dense tasks and achieves competitive performance to state-of-the-art methods even without extensive hyperparameter tuning. In particular, for an incremental prediction task, our method can reach its peak performance with much fewer events compared to the other methods (Fig. 11).  On the other hand, our method requires orders of magnitude less encoding computation compared to the existing methods. We also see that scaling up the memory dimensions improves performance on more complex tasks. Ultimately, we stress that the preliminary analysis presented here mostly shows the feasibility of using our representation in complex (dense) tasks; further architectural innovations are required to improve the performance of our model in these tasks. For example, existing architectures for such complex tasks are specially designed for structure-based processing (such as convolution on a 2D grid, 3D voxel, or graph) which differs from our unstructured encoding pipeline.
> >
> > >At what point does this become prohibitively expensive? 32 x 32 seems fairly small for a learned representation.
> >
> > Reply: Based on the reviewer's query, we perform an exhaustive search on the memory dimensions and their associated compute cost which reveals that (Figure 12 in page 18) even with a memory dimension of 2048x2048, an average of 210 MFLOPs/ev is required to encode events into the memory representation, which is still 10x smaller compared to the existing convolution-based dense encoding methods (25000 MFLOPs/ev). We believe that for such dense tasks, the overall computations will be mostly dominated by the dense-decoder block. We leave the development of a highly compute-efficient decoder block as a potential future work.
> >
> > >The outline of the method mostly is clear, but the Read, Write, and Erase sections are a little hard to follow. While the other parts are clearly outlined in equations, these three are condensed entirely in text under 'Associative memory augmented recurrent module', and it's not clear what goes where. Some clarifications here could help the reader.
> >
> > Reply: Thank you for pointing out this issue. Although we were unable to add any more details in the main draft due to space constraints, we have added a new section in Appendix I where we now elaborate on the details of the Read, Write, and Erase sections.
> >
> > >Table 3: AEGNN slightly outperforms the proposed method for N-Cars and should be bolded.
> >
> > Reply: Thank you for pointing this out. We have bolded the AEGNN N-Cars result accordingly in the revised manuscript.
> >
> > We appreciate the reviewer’s feedback which helped us to improve the quality of the paper. We also hope that our response and the revised paper have sufficiently addressed the reviewer’s concerns. We thank the reviewer again for their time and effort to improve the quality of our paper and look forward to further interactions.

---

> > > ### Author Response · Authors · 2022-11-24
> > > **Following up on the rebuttal - any further discussion on our current draft**
> > >
> > > Dear Reviewer rsty,
> > >
> > > We want to thank you again for your valuable time and feedback to improve the quality of the paper. We have tried our best to address most of your concerns during the rebuttal period. We would really appreciate it if you could please give us some feedback on whether or not our response has addressed your concerns. We will be more than happy to further discuss this with you if you have any other concerns regarding our current draft and the rebuttal. Please let us know about it!
> > >
> > > Thanks.
> > >
> > > Paper 4974 authors.

---

### Official Review · Reviewer_AhYC · 2022-10-26

**Confidence:** 4
**Correctness:** 4
**Technical Novelty And Significance:** 3
**Empirical Novelty And Significance:** 3
**Recommendation:** 8

**Clarity, Quality, Novelty And Reproducibility:**

I really enjoyed the quality of the illustrations, and the literature review is very detailed and gives the reader a good overview of the existing approaches to event encoding. Maybe the review section could benefit from more diversity, especially among papers that tackle egomotion, motion segmentation and depth - for example
 - Learning visual motion segmentation using event surfaces (3D, graph representation)
 - https://arxiv.org/pdf/1903.07520.pdf (dense image-like representation)
 - https://arxiv.org/pdf/1803.04523.pdf (image-like representation, classic pipeline)

In regards to novelty, I think the paper presents a relatively new (at least, to the field) problem and addresses it directly, while most of the existing papers to not consider the problem of event encoding as an independent one.

**Strength And Weaknesses:**

The paper presents and efficient way to encode and incrementally update the event stream, which is a pressing issue for any event-based processing pipeline; I believe the method could have a big impact if made publicly available to the community.

The evaluation is performed on N-Caltech101 and N-Cars for recognition tasks. While valid, the event camera (specifically - spatio-temporal event relations) emphasizes motion over the appearance, and an evaluation on any egomotion dataset or at least a more challenging recognition / segmentation would strengthen the paper.
We suggest for the future work (or, a revision of this paper if feasible) authors to add EVIMO (https://better-flow.github.io/evimo/) dataset to evaluation - egomotion, segmentation and depth estimation are all valuable applications for event cameras. MVSEC (https://daniilidis-group.github.io/mvsec/) is another
great source of data, with both egomotion and depth.


**Summary Of The Paper:**

The paper presents a high performance asynchronous encoding scheme for event-based camera data, which uses associative memory and preserves the spatio-temporal relationships between events in a compact and efficient manner. The authors evaluate the pipeline on recognition tasks, and achieve high speedups without sacrificing the accuracy of the predictions.


**Summary Of The Review:**

The asynchronous frame-less nature of the event stream is a big problem for event-based processing, and most modern methods either sacrifice information contained in the events, or the performance. I believe the paper has a potential to improve on the current state of the art, at least in terms of the encoding of the events. The only major weakness I see is the lack of focus / evaluation on depth and motion, with motion estimation (egomotion or segmentation) being the major applications of the event camera. But, the approach is still valuable and I would be willing to accept this paper provided the authors make their method available.

---

> ### Author Response · Authors · 2022-11-18
> **Response to the Reviewer AhYC**
>
> We are grateful to the reviewer for their positive feedback about our work. Below we try to address each of your comments:
>
> >We suggest for the future work (or, a revision of this paper if feasible) authors to add EVIMO (https://better-flow.github.io/evimo/) dataset to evaluation - egomotion, segmentation and depth estimation are all valuable applications for event cameras.
>
> Reply: We thank the reviewer for this valuable suggestion. The primary focus of this work was to develop an event-based representation learning framework and study the effectiveness of our representation on classification tasks. We considered extension to more complex tasks as a future research. However, based on all reviewers’ advice, we have performed preliminary experiments to study the feasibility of applying our method to more complex tasks that generate dense outputs. In particular, we developed a model architecture using our representation for motion segmentation using the EV-IMO dataset. However, due to the limitation of time, we experimented with only a subset of the EV-IMO dataset. The preliminary results (detailed in Appendix G) are shown below:
>
> |          Method                              |  IoU  |       | MFLOPs/ev (Encoder)|
> |:------------------------------------:|:---------:|---------|:-----------------------------:|
> |                                              | boxes | floor   |                                     |
> | EV-IMO                                | 0.70    | 0.59   | 4786.7                         |
> | EVDodgeNet                        | 0.67    | 0.61   | 2575.0                         |
> | SpikeMS                               | 0.59    | 0.46   | 18.5                             |
> |  EventFormer (m=64,d=32)  | 0.47    | 0.44   | 0.064                           |
> |  EventFormer (m=64,d=64)  | 0.49    | 0.43   | 0.193                           |
> | EventFormer (m=256,d=32) | 0.55    | 0.53   | 0.160                           |
>
> We observe that our event-based representation can be extended for dense tasks and achieves competitive performance to state-of-the-art methods even without extensive hyperparameter tuning. In particular, for an incremental prediction task, our method can reach its peak performance with much fewer events compared to the other methods (Fig. 11).  On the other hand, our method requires orders of magnitude less encoding computation compared to the existing methods. We also see that scaling up the memory dimensions improves performance on more complex tasks. Ultimately, we stress that the preliminary analysis presented here mostly shows the feasibility of using our representation in complex (dense) tasks; further architectural innovations are required to improve the performance of our model in these tasks. For example, existing architectures for such complex tasks are specially designed for structure-based processing (such as convolution on a 2D grid, 3D voxel, or graph) which differs from our unstructured encoding pipeline.
>
> >I really enjoyed the quality of the illustrations, and the literature review is very detailed and gives the reader a good overview of the existing approaches to event encoding.
>
> Reply: We are really grateful to the reviewer for such positive feedback.
>
> >Maybe the review section could benefit from more diversity, especially among papers that tackle egomotion, motion segmentation, and depth
>
> Reply: Thank you for this advice. We have modified the review section by adding a brief discussion (due to the space constraints) on the recent works which focus on solving egomotion, motion segmentation, and depth-estimation tasks directly from event-camera data. Please let us know if you have any further suggestions on this, we will modify it in the final version accordingly.
>
> >But, the approach is still valuable and I would be willing to accept this paper provided the authors make their method available.
>
> Thank you again for the positive feedback. Indeed, we are working on a well-documented version of our codebase which, we believe, will be more useful to the community and we hope to make the method available by the time we receive the final decision.
>
> We thank the reviewer again for their time and effort to improve the quality of our paper.

---

> > ### Author Response · Authors · 2022-11-24
> > **Following up on the rebuttal - any further discussion on our current draft**
> >
> > Dear Reviewer AhYC,
> >
> > We want to thank you again for your valuable time and feedback to improve the quality of the paper. We have tried our best to address most of your concerns during the rebuttal period. We would really appreciate it if you could please give us some feedback on whether or not our response has addressed your concerns. We will be more than happy to further discuss this with you if you have any other concerns regarding our current draft and the rebuttal. Please let us know about it!
> >
> > Thanks.
> >
> > Paper 4974 authors.

---

### Official Review · Reviewer_rHjG · 2022-10-27

**Confidence:** 4
**Correctness:** 3
**Technical Novelty And Significance:** 3
**Empirical Novelty And Significance:** 3
**Recommendation:** 6

**Clarity, Quality, Novelty And Reproducibility:**

The paper is well written and the hybrid transformer and rnn design has certain novelty.

**Strength And Weaknesses:**

Pros:
1. This paper is well written and clear in general.
2. The proposed EventFormer seems to perform well on two event-based object recognition datasets.
3. The hybrid transformer and rnn architecture design seems novel.

Cons:
1. The input of EventFormer is a list of events {(x_i, y_i)}, does it mean that the polarity information of events is lost? Does it have any influence to the performance?
2. As is mentioned in paper, the MFLOPs/ev on average window of 25000 events are reported. However, 50ms long sequence is used for each sample in supplementary material. What exactly does that mean?
3. The resolution of N-Caltech101 should be 180×240, but the paper says 180×260, is this a typo or is there pre-processing?
4. How is the efficiency of the proposed method when dealing with high spatial resolution events such as the ones produced by Prophesee Gen4? This is especially important for tasks such as object detection, semantic segmentation and optical flow estimation.
5. Authors are suggested to provide inference speed other than flops of models.
6. What about performance of the proposed method on other tasks such as object detection, depth estimation and optical flow estimation?


**Summary Of The Paper:**

This paper introduces a new event representation framework EventFormer, which can process sparse events asynchronously and efficiently. The author proposes a memory-based mechanism to retrieve the past representation, store the memory with new representation and update the hidden states of associative memory. The paper claims that the method obtains high performance with minimal compute cost on the N-Caltech101 and N-Cars dataset.

**Summary Of The Review:**

Overall, I think the design is interesting and seems to work to some extent. However, authors are suggest to apply the proposed representation on more advanced tasks such as object detection and optical flow estimation instead of only classification.

---

> ### Author Response · Authors · 2022-11-18
> **Response to the Reviewer rHjG 1/2**
>
> We are glad to know that the reviewer considers our method interesting. Below we try to address each of your comments:
>
> > The input of EventFormer is a list of events {(x_i, y_i)}, does it mean that the polarity information of events is lost? Does it have any influence to the performance?
>
> Reply: Thank you for bringing this up. In our current formulation, we did not consider polarity information of the events during the event-encoding part. According to your advice, we have conducted an additional experiment that now considers polarity as an additional input alongside the positional coordinates of the events and included the results in Appendix F. We observe that adding polarity information has a very negligible influence ($0.849$ compared to the original $0.848$) on the overall performance.
>
> > As is mentioned in paper, the MFLOPs/ev on average window of 25000 events are reported. However, 50ms long sequence is used for each sample in supplementary material. What exactly does that mean?
>
> Reply: We visually observed (using aggregated event-frames) that, for both NCars and N-Caltech101 datasets, a sequence length of 40-50ms was able to capture sufficient structural information of the target object. Therefore, we used 50ms long sequence for each sample for both training and testing. On the other hand, we used a fixed (not average) window of 25000 events to calculate the average MFLOPs/ev; this approach has been used in prior papers for reporting computational cost, and hence, we have used the same to make a fair comparison. We have corrected the statement (please see on Page 9, the blue highlighted sentence below table 3) in the revised manuscript.
>
> >The resolution of N-Caltech101 should be 180×240, but the paper says 180×260, is this a typo, or is there pre-processing?
>
> Reply: We apologize for the typo; the resolution of N-Caltech101 was indeed 180x240. We have corrected the typo in the revised draft.
>
> >How is the efficiency of the proposed method when dealing with high spatial resolution events such as the ones produced by Prophesee Gen4? This is especially important for tasks such as object detection, semantic segmentation, and optical flow estimation.
>
> Reply: We thank the reviewer for raising this important issue. We could not experiment directly on Prophesee Gen4 dataset as the task (object detection) considered in this dataset as extending our representation for object detection require major advancements, and was not feasible during the rebuttal period. However, we agree that understanding the computational complexity and performance of our model for more complex tasks and/or higher resolution is important. Therefore, we have presented additional experiments on a relatively more complex (dense) task, namely, motion segmentation, using a higher spatial resolution of the input (256x256). Our preliminary results (Table 5 in Appendix G) show that a larger $m$ significantly improves accuracy while a larger $d$ has relatively less impact. In other words, we see increasing $m$ is more important than increasing $d$ to learn an effective representation from higher spatial resolution events. Furthermore, the computational complexity analysis reveals that (Figure 12 in page 18) even with a memory dimension of 2048x2048, an average of 210 MFLOPs/ev is required to encode events into the memory representation, which is still 10x smaller compared to the existing dense encoding methods (25000 MFLOPs/ev).
>
> >Authors are suggested to provide inference speed other than flops of models.
>
> Reply: We thank the reviewer for this suggestion. We compared the inference speed of our method with the most efficient asynchronous method (AEGNN) (‘Computation Latency’ section on Page 9) for classification tasks by mapping (running) both methods in the same platform (PyTorch implementation on an NVIDIA-RTX3090 GPU). Further, in Appendix H, we have also added a timing comparison between EventFormer and the best-performing dense method (Matrix-LSTM). In both of the preceding cases, we implemented all models (AEGNN, Matrix-LSTM, and EventFormer) in the same machine while comparing latency. However, a direct comparison of the latency of ours and other methods in Table 3 is challenging as different methods have provided their latency measurements using different hardware configurations and hyperparameters. However, we still added Table 7 in Appendix J (page 19) showing the latency reported in the original papers for the methods and corresponding hardware configuration. A common methodology was used in all these works where the average latency/event was measured considering the average run time per sample divided by the average number of events per sample.
> If accepted, we will explore the feasibility of implementing many of these methods (provided the availability of the codes for each model) in our hardware configuration to report a complete latency comparison in the final version of the draft.

---

> > ### Author Response · Authors · 2022-11-18
> > **Response to the Reviewer rHjG 2/2**
> >
> > >What about the performance of the proposed method on other tasks such as object detection, depth estimation, and optical flow estimation?
> >
> > Reply: We thank the reviewer for this important question. The primary focus of this work was to develop an event-based representation learning framework and study the effectiveness of our representation on classification tasks. We considered an extension to more complex tasks as a future research. However, based on all reviewers’ advice, we have performed preliminary experiments to study the feasibility of applying our method to more complex tasks that generate dense outputs. In particular, we developed a model architecture using our representation for motion segmentation using the EV-IMO dataset. However, due to the limitation of time, we experimented with only a subset of EV-IMO dataset. The preliminary results (detailed in Appendix G) are shown below:
> >
> > |          Method                              |  IoU  |       | MFLOPs/ev (Encoder)|
> > |:------------------------------------:|:---------:|---------|:-----------------------------:|
> > |                                              | boxes | floor   |                                     |
> > | EV-IMO                                | 0.70    | 0.59   | 4786.7                         |
> > | EVDodgeNet                        | 0.67    | 0.61   | 2575.0                         |
> > | SpikeMS                               | 0.59    | 0.46   | 18.5                             |
> > |  EventFormer (m=64,d=32)  | 0.47    | 0.44   | 0.064                           |
> > |  EventFormer (m=64,d=64)  | 0.49    | 0.43   | 0.193                           |
> > | EventFormer (m=256,d=32) | 0.55    | 0.53   | 0.160                           |
> >
> > We observe that our event-based representation can be extended for dense tasks and achieves competitive performance to the state-of-the-art methods even without extensive hyperparameter tuning. In particular, for an incremental prediction task, our method can reach its peak performance with much fewer events compared to the other methods (Fig. 11).  On the other hand, our method requires orders of magnitude less encoding computation compared to the existing methods. We also see that scaling up the memory dimensions improves performance on more complex tasks. Ultimately, we stress that the preliminary analysis presented here mostly shows the feasibility of using our representation in complex (dense) tasks; further architectural innovations are required to improve the performance of our model in these tasks. For example, existing architectures for such complex tasks are specially designed for structure-based processing (such as convolution on a 2D grid, 3D voxel, or graph) which differs from our unstructured encoding pipeline.
> >
> > We appreciate the reviewer’s feedback which helped us to improve the quality of the paper. We also hope that our response and the revised paper have sufficiently addressed the reviewer’s concerns. We thank the reviewer again for their time and effort to improve the quality of our paper and look forward to further interactions.

---

> > > ### Author Response · Authors · 2022-11-24
> > > **Following up on the rebuttal - any further discussion on our current draft**
> > >
> > > Dear Reviewer rHjG,
> > >
> > > We want to thank you again for your valuable time and feedback to improve the quality of the paper. We have tried our best to address most of your concerns during the rebuttal period. We would really appreciate it if you could please give us some feedback on whether or not our response has addressed your concerns. We will be more than happy to further discuss this with you if you have any other concerns regarding our current draft and the rebuttal. Please let us know about it!
> > >
> > > Thanks.
> > >
> > > Paper 4974 authors.

---

### Official Review · Reviewer_wCjB · 2022-10-28

**Confidence:** 3
**Correctness:** 4
**Technical Novelty And Significance:** 3
**Empirical Novelty And Significance:** 4
**Recommendation:** 6

**Clarity, Quality, Novelty And Reproducibility:**

this paper is clearly written, but with some typos. The implementation details are enough for re-implementation.

**Strength And Weaknesses:**

Strength
1. the idea to adopt the Transformer and memory scheme for event-based representation learning is interesting and new for this problem.
2. the model works well on short-term and simple classification datasets, including n-caltech101, and n-cars.


Weaknesses
1. the writing of this paper still needs further improvements, many typos can be found, even in the figure;
2. the performance on the long-term event streams, and challenging tasks is not verified.

**Summary Of The Paper:**

A memory-augmented representation learning model, EventFormer, is proposed for asynchronous event-based perception. To achieve this goal, the EventFormer learns to store, retrieve and update its memory representation in the latent form of higher-order spatiotemporal dynamics of the events. The idea is interesting and also obtains good performance on two classification tasks. As the drawbacks of this paper, many typos can be found in the manuscript. I also wonder how about the performance on the long-term event representation learning and also other challenging tasks, such as event-based object detection, event-based visual tracking, etc.

**Summary Of The Review:**

A memory-augmented representation learning model, EventFormer, is proposed for asynchronous event-based perception. To achieve this goal, the EventFormer learns to store, retrieve and update its memory representation in the latent form of higher-order spatiotemporal dynamics of the events. The idea is interesting and also obtains good performance on two classification tasks.
The following issues can be addressed in future versions to further improve the quality of this work.
1. the writing of this paper still needs further improvements, many typos can be found, even in the figure;
2. the performance on the long-term event streams, and challenging tasks is not verified.

---

> ### Author Response · Authors · 2022-11-18
> **Response to Reviewer wCjB**
>
> We are glad to know that the reviewer considers our method new and interesting. Below we try to address each of your comments:
>
> >The writing of this paper still needs further improvements, many typos can be found, even in the figure.
>
> Reply:  We are sorry for the typos that can be found in the paper. We have carefully revised the manuscript and corrected the typos in the text (highlighted in blue color (words)) and figure (e.g. Figure 1(a)). We will continue to revise the paper to rectify any further typos.
>
> >The performance on the long-term event streams, and challenging tasks is not verified.
>
> Reply: The primary focus of this paper was to develop an event-based representation learning framework and study the effectiveness of our representation on classification tasks. We considered an extension to more complex tasks as future research. However, based on all reviewers’ advice, we have performed preliminary experiments to study the feasibility of applying our method to more complex tasks that generate dense outputs. In particular, we developed a model architecture using our representation for motion segmentation using the EV-IMO dataset. However, due to the limitation of time, we experimented with only a subset of the EV-IMO dataset. The preliminary results (detailed in Appendix G) are shown below:
>
> |          Method                              |  IoU  |       | MFLOPs/ev (Encoder)|
> |:------------------------------------:|:---------:|---------|:-----------------------------:|
> |                                              | boxes | floor   |                                     |
> | EV-IMO                                | 0.70    | 0.59   | 4786.7                         |
> | EVDodgeNet                        | 0.67    | 0.61   | 2575.0                         |
> | SpikeMS                               | 0.59    | 0.46   | 18.5                             |
> |  EventFormer (m=64,d=32)  | 0.47    | 0.44   | 0.064                           |
> |  EventFormer (m=64,d=64)  | 0.49    | 0.43   | 0.193                           |
> | EventFormer (m=256,d=32) | 0.55    | 0.53   | 0.160                           |
>
> We observe that our event-based representation can be extended for dense tasks and achieves competitive performance to the state-of-the-art methods even without extensive hyperparameter tuning. In particular, for an incremental prediction task, our method can reach its peak performance with much fewer events compared to the other methods (Fig. 11 on Page 17).  On the other hand, our method requires orders of magnitude less encoding computation compared to the existing methods. We also see that scaling up the memory dimensions improves performance on more complex tasks. Ultimately, we stress that the preliminary analysis presented here mostly shows the feasibility of using our representation in complex (dense) tasks; further architectural innovations are required to improve the performance of our model in these tasks. For instance, existing architectures for such complex tasks are specially designed for structure-based processing (such as convolution on a 2D grid, 3D voxel, or graph) which differs from our unstructured encoding pipeline.
>
> We do agree with the reviewer that we did not evaluate our method on long-term event streams as the standard datasets that we used have relatively short-term event sequences. In our future work, we plan to evaluate our method on more complex tasks (such as event-based object tracking during a long period of time) that would involve longer-event sequences.
>
> We appreciate the reviewer’s feedback which helped us to improve the quality of the paper. We also hope that our response and the revised paper addressed the reviewer’s concerns. We thank the reviewer again for their time and effort to improve the quality of our paper and look forward to further interactions.

---

> > ### Author Response · Authors · 2022-11-24
> > **Following up on the rebuttal - any further discussion on our current draft**
> >
> > Dear Reviewer wCjB,
> >
> > We want to thank you again for your valuable time and feedback to improve the quality of the paper. We have tried our best to address most of your concerns during the rebuttal period. We would really appreciate it if you could please give us some feedback on whether or not our response has addressed your concerns. We will be more than happy to further discuss this with you if you have any other concerns regarding our current draft and the rebuttal. Please let us know about it!
> >
> > Thanks.
> >
> > Paper 4974 authors.

---

### Author Response · Authors · 2022-11-18
**Summary of the Revision**

We are grateful to all the reviewers for their time and helpful feedback to improve the quality of our work. Based on their suggestions, we have made the following changes to the main paper:

- Rectified several typos throughout the paper.
- Diversified the review section by adding a brief discussion on the relevant works that tackled more challenging tasks other than classification.
- Added a new section: 'Possible Scaling for More Complex Tasks', to outline the potential application of our representation learning framework on more challenging tasks.

In the appendix, we have made the following changes:

- A new Appendix F is included to study the impact of adding polarity information in our framework.
- A new Appendix G is included to conduct a feasibility study of applying our method to complex tasks. We provide several important insights and possible directions for future research on how to scale up our method for such complex tasks through extensive quantitative and qualitative experiments.
- A new Appendix H is included to study how our latency would scale up for an increasing number of events.
- A new Appendix I is included to elaborate on the inner workings of the memory operators (Read, Write, and, Erase).
- A new Appendix J is included to report the latency measurements of other methods.

---

### Author Response · Authors · 2022-12-06
**A gentle reminder for author-reviewer discussion**

Dear reviewers,

We want to thank you once again for your valuable time and constructive feedback to improve the quality of the work. Since the 2nd-phase of the discussion period will end after this week, we would really appreciate it if we could hear your comments on our revised paper. If our responses have addressed the majority of your concerns, we would really appreciate it if you could reconsider your scores. Also, we will be happy to answer any further queries that you may have, please let us know about them.

Thanks.

The Paper4974 Authors

---

### Decision · Program_Chairs · 2023-01-20

**Decision:**

Accept: notable-top-25%

**Justification For Why Not Higher Score:**

Interesting work but I am just not sure it would be good for an oral.

**Justification For Why Not Lower Score:**

N/A

**Metareview: Summary, Strengths And Weaknesses:**

The paper presents EventFormer, a representation learning framework for asynchronous spatiotemporal representation that uses external associative memory for efficient event-based processing.
The authors revised the paper to address all major concerns -- especially those regarding timing --  of the reviewers and included a series of appendices to further strengthen their contribution.
The idea is novel and I like the approach which, despite few limitations, will make a very interesting addition to the conference.

**Note From Pc:**

if the above contains the word "oral" or "spotlight" please see: "oral" presentation means -> notable-top-5% and "spotlight" means -> notable-top-25%. As stated in our emails, we are disassociating presentation type from AC recommendations